# Validation of Pharmacogenomic Interaction Probability (PIP) Scores in Predicting Drug–Gene, Drug–Drug–Gene, and Drug–Gene–Gene Interaction Risks in a Large Patient Population

**DOI:** 10.3390/jpm12121972

**Published:** 2022-11-29

**Authors:** Kristine Ashcraft, Kendra Grande, Sara L. Bristow, Nicolas Moyer, Tara Schmidlen, Chad Moretz, Jennifer A. Wick, Burns C. Blaxall

**Affiliations:** 1Invitae Corporation, 1400 16th Street, San Francisco, CA 94103, USA; 2The Christ Hospital Health Network, 2139 Auburn Avenue, Cincinnati, OH 45219, USA

**Keywords:** pharmacogenomics, population health, adverse drug events, phenoconversion, pharmacogenetics, precision medicine, medication management

## Abstract

Utilizing pharmacogenomic (PGx) testing and integrating evidence-based guidance in drug therapy enables an improved treatment response and decreases the occurrence of adverse drug events. We conducted a retrospective analysis to validate the YouScript^®^ PGx interaction probability (PIP) algorithm, which predicts patients for whom PGx testing would identify one or more evidence-based, actionable drug–gene, drug–drug–gene, or drug–gene–gene interactions (EADGIs). PIP scores generated for 36,511 patients were assessed according to the results of PGx multigene panel testing. PIP scores versus the proportion of patients in whom at least one EADGI was found were 22.4% vs. 22.4% (*p* = 1.000), 23.5% vs. 23.4% (*p* = 0.6895), 30.9% vs. 29.4% (*p* = 0.0667), and 27.3% vs. 26.4% (*p* = 0.3583) for patients tested with a minimum of 3-, 5-, 14-, and 25-gene panels, respectively. These data suggest a striking concordance between the PIP scores and the EAGDIs found by gene panel testing. The ability to identify patients most likely to benefit from PGx testing has the potential to reduce health care costs, enable patient access to personalized medicine, and ultimately improve drug efficacy and safety.

## 1. Introduction

With the introduction of the Right Drug Dose Now Act in the United States Congress [1], in order to address the barriers into integrating pharmacogenomic (PGx) testing into clinical practices, it is clear that PGx is becoming a priority for informing drug therapy decisions. One of the major goals of this new legislation is to reduce adverse drug events caused by evidence based, actionable drug–gene interactions (EADGIs), which for the purposes of this study are drug–gene interactions (DGIs), drug–drug–gene interactions (DDGIs), and drug–gene–gene interactions (DGGIs), with the drug or dose change guidance determined by the U.S. Food and Drug Administration (FDA) or the Clinical Pharmacogenetics Implementation Consortium (CPIC) category A or B guidelines [2,3]. Meeting this goal necessitates the use of genotyping and evidence-based PGx guidelines to predict patients’ responses to medications and toxicity risks [2,4,5]. Panel-based PGx testing can be used in concert with a clinical decision support tool (CDST) to guide treatment, either contemporaneously or as new indications arise [6]. Preemptive PGx has the potential to enable informed decisions for many patients [7], as PGx variants, resulting in an atypical drug response, are highly prevalent, exceeding 99% [8,9]. However, the high cost and the lack of infrastructure both remain barriers to the preemptive testing of all patients [10].

Automated algorithms that identify potential EADGI risks have shown a utility in determining which patients would benefit most from PGx testing [11,12,13]. Invitae’s CDST YouScript, uses a patented PGx interaction probability (PIP) score to predict how likely a patient is to have an EADGI [14]. The PIP score highlights patients most likely to benefit from PGx testing based on their medications and the prevalence of population-based phenotypes that may result in an EADGI [11,13,15]. PGx-guided medication management for patients with medications that would increase the PIP score has resulted in a significant decrease in hospitalizations, emergency department visits, and healthcare costs in an observational study [11]. In a prospective randomized trial of patients aged 50 years and older, PGx profiling with CDST guidance in drug therapy showed reduced re-hospitalizations and emergency department visits when compared with medication management using a standard drug information resource, both study arms with pharmacist support [13]. For patients hospitalized with COVID-19, moderate (25–50%) or high PIP scores (>50%) with no PGx testing were associated with longer lengths of stay than those whose PIP scores were low (<25%), while risk adjustment factor (RAF) scores, a standard measure of patient complexity, were not associated with the length of stay [16]. 

As the incidence of complex diseases that require polypharmacy is increasing, assessing, and mitigating PGx risk can reduce the length of hospital stays, decrease the healthcare costs, and improve patient outcomes. The aim of this study is to further validate the accuracy of the PIP score in predicting patients who may have EADGIs and would benefit most from PGx testing. This retrospective analysis of approximately 36,000 patients compared the pretest PIP scores with the PGx multigene panel test results and highlighted the common risks detected.

## 2. Materials and Methods

### 2.1. Study Population

Individuals residing in the United States, who underwent provider-ordered PGx testing, at minimum for *CYP2C19, CYP2C9,* and *CYP2D6* through Invitae (Seattle, WA, USA previously Genelex) from May 2013 to March 2022, were included in the study. The specific PGx genes tested varied, ranging from 3 to 25 genes based on provider ordering preferences and the tests that were available at the lab. At the time of testing, patients must have had at least one medication reported on their requisition form or have had a current medication list sourced from the attached electronic health record, in order to allow for the calculation of PIP scores based on the medication regimens. A complete list of the drugs used to calculate the PIP scores in this study, and their respective clinical areas is available in the Appendix A. 

All patient data were de-identified and recorded in a Health Insurance Portability and Accountability Act (HIPAA)-compliant electronic database. Review and analysis of de-identified and aggregated data were approved for waiver of authorization by WCG IRB (Puyallup, WA, USA). 

### 2.2. YouScript Algorithm: Pharmacogenomic Interaction Probability

PIP scores were calculated for each patient as previously described [15,16]. Briefly, each PIP score was generated based on two sets of information: (1) the list of medications prescribed to the patient and (2) the prevalence of certain pharmacokinetic and pharmacodynamic phenotypes in the North American population. Race and ethnicity were not accounted for in the PIP scores due to the limitations in the self-reported patient data. In addition, age was not factored into calculating the PIP scores as this would not impact EADGI detection, unless a criterion, such as Beers was applied, and this optional factor was not included in this analysis. The PIP algorithm used this information to calculate the probability that one or more EADGIs with moderate, major, or contraindicated severity would be detected for each patient (Figure 1). Moderate EADGIs can result in substantial clinical effects and are associated with clinically actionable recommendations, such as adjusting a dose or prescribing an alternative drug. Major or contraindicated EADGIs are typically associated with drug change guidance or drug avoidance, when the risks of using a particular drug likely outweigh the benefits. Actionable recommendations are based on FDA labeling or CPIC guidelines category A or B [2,3]. 

Certain interactions were not included in the PIP score calculations, including interactions that required monitoring and interactions that would not increase the severity of an already detected drug–drug interaction because the added information from PGx testing would not change the clinical management. 

Importantly, phenoconversion, predicted by incorporating published evidence in changes in the area under the curve (AUC) for DGGIs and DDGIs, was accounted for in PIP scores. 

Another YouScript algorithm was used to determine the likelihood and severity of all potential EADGIs (i.e., DGIs, DDGIs, DGGIs) plus additional drug–drug interactions (DDIs) and drug–drug–drug interactions (DDDIs). 

### 2.3. Analysis

For each individual in the cohort, two PIP scores were calculated. First, a 25-gene PIP score was calculated using the medications listed for the patient at the time of testing. However, because the 25-gene PIP score did not necessarily reflect the genes selected by the clinicians, or the genes available at the time of testing, the second PIP score was an adjusted PIP score, as determined using only the PGx genes for which the patient received a laboratory test. After PGx testing, interpreted phenotypes and the presence of EADGIs were determined for each patient. 

To validate the performance of the PIP algorithm, the adjusted PIP scores were compared with the proportion of patients who were determined, by PGx testing, to have at least one EADGI (i.e., EADGI rate). This comparison was made in the overall cohort (i.e., 3–25 genes ordered) and in the subgroups of patients who had testing with a minimum set of 5, 14, or 25 genes. In the 5-gene group, patients had to have had at least the three base genes plus *CYP3A4* and *CYP3A5* ordered by their clinicians. In the 14-gene group, patients had to have had at least the genes in the 5-gene panel plus *CYP2B6, CYP4F2, SLCO1B1, TPMT, DPYD, HLA-B*57:01, IFNL3, UGT1A1,* and *VKORC1* ordered. The 25-gene panel included all genes in the 14-gene panel plus *ADRA2A, COMT, CYP1A2, F2, F5, GRIK4, HTR2A, HTR2C, MTHFR, NAT2,* and *OPRM1*. Of note, the PIP scores and presence of EADGIs were based on all the genes ordered from the 25 gene list and the listed medications, instead of being solely based on the minimum set of genes. 

For the entire cohort, the EADGIs determined by PGx testing were categorized according to the interaction level, estimated change in AUC, and the primary clinical area. The AUC change was based on the appropriate published literature for any given interaction. The primary clinical area was assigned according to the most commonly used indication for each drug, as determined by Invitae’s clinical PGx pharmacist team, utilizing patient-friendly headings with the Anatomical Therapeutic Chemical (ATC) classification system as a guide [17]. The number needed to test (NNT), or how many patients who would need to be tested to identify one EADGI, was calculated based on the number of patients found by PGx testing to have EADGIs.

The statistical significance of a single proportion, compared to a population estimate, was determined using the z-test with a two-sided alpha level of 5% [18]. A *p*-value of less than 0.05 indicated statistical significance, and a greater value indicated that no evidence of a significant difference was found. 

## 3. Results

### 3.1. Characterization of the Study Population

Among the 36,511 patients included in the cohort, 56.3% were female, 47.2% were white, and the mean age was 61 years (range, 0–110 years) (Table 1). The mean number of medications, including vitamins and supplements, per person was 9.4 (range, 1–62). Although more than three-quarters of patients (78.4%) had the 5-gene minimum gene panel ordered, only 8.7% and 5.7% of patients had the 14- or 25-gene panel ordered, respectively. The characteristics of individuals in these three subgroups were generally similar to those in the overall cohort; however, race and ethnicity were largely unknown in the 14- and 25-gene panel subgroups, and the mean age and the mean number of medications were lower in the 25-gene panel subgroup than in the others. The overall variant phenotype rate was high, with 100% of the tested patients having at least one variant detected when at least 14 genes were assessed.

Among all the patients, 30,471 (83%) were taking at least one high-impact PGx medication that could result in an EADGI (Figure 2). Almost 60% of patients were taking medications that impacted more than one clinical area. EADGIS were identified in the clinical areas of behavioral health, cardiology, pain management, hematology and oncology, infectious disease, gastroenterology, urology, transplant, reproductive and sexual health, neurology, rheumatology, endocrinology, and miscellaneous. Of note, 2270 patients (7.4%) were taking medications with a known PGx impact in behavioral health, pain management, and cardiology concurrently. 

### 3.2. Comparison of PIP Scores to PGx Test Results

Among the entire cohort (with a minimum of three genes tested), the mean PIP and the mean adjusted PIP scores were 26.4% and 22.4%, respectively, and 22.4% of patients (*p* = 1.0000) were determined by PGx testing to have had at least one EADGI (Table 2). For a subset of patients, 65 years and older (*n* = 18,124), the mean PIP and the mean adjusted PIP scores were 32.6% and 27.3%, respectively, and 27.3% of the patients (*p* = 1.0000) were determined by PGx testing to have at least one EADGI. Notably, 11,861 patients (32.5%) had an adjusted PIP score of 0.0%, which indicated that they were taking no medications with a high-evidence PGx impact at the time of testing. When these patients were appropriately removed from the analysis, the mean PIP score among the remaining 24,650 patients rose to 33.2%, and 33.2% of the patients (*p* = 1.0000) were found to have had at least one EADGI, resulting in an NNT of 3.0. A total of 17,703 patients had an adjusted PIP score of >25%, which based on the previous studies, indicated a moderate (26–50%) to high (>50%) likelihood of an EADGI being detected [15,16]. Among the subgroups of patients, who had testing with a minimum of 5, 14, and 25 genes, no significant differences were observed between mean adjusted PIP scores and EADGI rates (Table 2). 

### 3.3. Characterization of All Interactions

Figure 3 shows all interactions stratified according to type and severity per patient. The percentage of patients, with at least one interaction, increased as the adjusted PIP score threshold increased. A moderate-risk PIP score of >25% is the current minimum in the two studies that are in process, which utilize the PIP score. Patients with DDGIs or DGGIs (i.e., phenoconversion) comprised 5.9% of all patients and 14.7% of all patients with an EADGI. For these patients, the mean adjusted PIP score was 43.2%, with 43.3% of the patients (*p* = 0.7882) having at least one EADGI. 

A total of 9804 EADGIs were detected in 9134 patients; 2444 (24.9%) of these were major or contraindicated interactions. Table 3 shows the breakdown of EADGIs by level of severity, according to the clinical area with the most commonly used indication for each drug. Behavioral health had the highest proportion of major or contraindicated EADGIs (45.3%) followed by cardiology (35.9%) and pain management (18.7%).

Medications across multiple clinical areas were typically the cause of DDGIs involving a cumulative impact on the AUC (Table 4), thus causing a greater change in drug exposure than in binary interactions.

Medications found to have EADGIs were ranked according to the frequency with which they were prescribed within a calendar year, as determined by the 2019 ClinCalc DrugStats Database [19]. The top five drugs with EADGIs are shown in Table 5, and the top five drugs with major or contraindicated EADGIs are shown in Table 6. Codeine and tramadol were two of the medications most commonly involved in major EADGIs. Clopidogrel was the second most common medication to have a major EADGI, while es (citalopram) was most likely to flag a major EADGI (Table 6). 

## 4. Discussion

The results from this study, of more than 36,000 patients, demonstrate that the PIP score is an accurate predictor of how frequently EADGIs are detected, based on the comparison with PGx testing results. Importantly, the accuracy of the adjusted PIP scores to PGx test results was comparable regardless of the number of minimum genes tested. Taken together, these data demonstrate that the use of a CDST, with an embedded PIP score calculator, to identify patients who would benefit most from PGx testing is a practicable alternative to universal preemptive testing. PGx-guided medication management has the potential to improve the safety and efficacy of drugs and reduce inappropriate use of multiple medications for all patients [20]. In fact, the prevalence of variants in our cohort ranged from 97% to 100%, similar to larger studies indicating that >99% of the population carries at least one genetic variant that results in an atypical response to at least one medication [8,21]. As health care institutions and payers consider the benefits of PGx-guided prescribing, streamlining costs and resources remains a priority. In a prior study, comparing patient identification via manual review versus utilization of the PIP algorithm, the number of patients identified as being likely to test positive for a clinically significant PGx interaction was increased by over two-fold [15].

The findings from this study also demonstrate the importance of implementing PGx-guided medication management with a more global approach. Traditionally, reporting has been limited to a single or a few genes, based on a test order (given that reimbursement for PGx testing has historically been for single drug–gene pairs), to a panel targeted to a specific clinical area, such as behavioral health, or to limited conditions, such as major depressive disorders. This trend is consistent in this cohort, with only 8.7% and 5.7% of patients having the 14- or 25-gene panel ordered, respectively. However, many PGx testing laboratories typically run multigene panels, as high-throughput assays can reduce the overall cost and turnaround times without compromising sensitivity and specificity [22]. Unfortunately, most commercial payers have not followed suit even though PGx panels provide more actionable information that can lead to an improved medication management and to reduced adverse outcomes. Another limitation of the current ordering and reporting practices is that only EADGIs, for a single clinical area, are reported despite the potential impact in other clinical areas. In this study, the most common clinical area crossover, for drugs with EADGIs, was observed for behavioral health, pain management, and cardiology, and 7.4% of the patients tested were taking at least one medication with a known EADGIs in each of these clinical areas. By reporting on only one indication or clinical area, the potential PGx impact of medications, prescribed either contemporaneously or in the future, may be missed. Policies that influence laboratories to report only partial results create potentially avoidable morbidity and mortality risks, raising serious ethical concerns [23]. Furthermore, because some payers will only reimburse one panel test for the lifetime of a patient, the broadest possible testing should be considered to optimize lifelong benefit.

For the 25% of EADGIs detected in our study that were major or contraindicated, a different medication would have been recommended because the potential risk outweighed the benefit. The the top five medications, with major or contraindicated EADGIs, were es (citalopram), clopidogrel, tramadol, and codeine. The PIP score for a patient simultaneously taking es (citalopram), clopidogrel, and codeine/tramadol was 62%, meaning an EADGI would be identified more often than not at the time of testing. 

Importantly, many of these medications have interactions with the same gene but are used to treat across different clinical areas. For example, *CYP2C19* has established EADGIs with 21 medications used in behavioral health, cardiology, gastroenterology, infectious disease, neurology, pain management, and reproductive and sexual health. This includes es (citalopram) (PIP score, 32%; NNT, 3.1) and clopidogrel (PIP score, 29%; NNT, 3.4). It was reported that patients with the most extreme *CYP2C19* metabolizer phenotypes were increased by approximately 9% in suicide victims taking es (citalopram) when compared with a control population [24]. Another study in children using es (citalopram) showed that testing 463 patients would prevent one suicide or an attempt [25]. Both illustrate the potential serious clinical repercussions of using disease-specific PGx medication guidance instead of a comprehensive PGx-guided medication management. Similarly, it was estimated in a 1000-patient cohort that if all patients had *CYP2C19* genotyping, to guide clopidogrel prescribing over a one-year period, 20 deaths due to cardiovascular disease would be averted—a 27% decrease, or one life saved for every 50 patients tested [26]. Despite this, PGx testing for patients on clopidogrel is not yet routine, even though in 2010 the FDA issued a boxed warning for clopidogrel regarding the dangers for patients with *CYP2C19* variants [27]. This highlights safety concerns with clinicians ordering PGx testing for *CYP2C19* to inform the prescribing of behavioral health medications, without considering other interactions, which could have an impact on medications prescribed (by other providers) for other indications (e.g., clopidogrel). A recent study of the impact of a behavioral health-focused PGx panel test in veterans showed a faster time to remission of major depressive disorders and a reduction in the medications that would cause a known EADGI in behavioral health treatments; however, no mention was made of applying the information for EADGIs outside of that clinical area of behavioral health [7]. Similarly, codeine and tramadol (for each medication: PIP score, 9%; NNT, 11) for pain management both have known interactions with *CYP2D6*. However, *CYP2D6* currently has 51 EADGIs for medications used in behavioral health, cardiology, gastroenterology, oncology and hematology, infectious disease, neurology, reproductive and sexual health, and urology. In 2017, the U.S. President officially declared the opioid crisis a public health emergency [28], and in 2021, CPIC published guidelines for personalizing many opioids, primarily based on *CYP2D6* [29]. When patients in opioid dependency treatment had PGx testing, more than 20% were found to be *CYP2D6* poor or ultrarapid metabolizers, an approximately three-fold higher frequency than in a typical U.S. population [30,31]. A unmanaged pharmacogenomic risk is likely impacting 8–12% of people using opioids for chronic pain who then develop an opioid use disorder [32]. 

With a mean medication count of 9.4 in the cohort, co-medication is also of concern. Importantly, PIP scores account for phenoconversions, specifically those caused when pharmacokinetic drug interactions amplify or attenuate an individual’s inherent genetic ability to metabolize medications [33,34]. Drug-mediated phenoconversion was shown to impact one in four patients treated with PGx medications while taking an inhibitor or inducer of the same enzyme [35]. Similarly, the impact of phenoconversion on PGx cytochrome P450 (CYP) metabolism was shown to change the phenotype and consequently the clinical management of approximately 15% of acute care psychiatric inpatients [36]. Of the EADGIs that were the 10 most common phenoconversions detected, with a cumulative impact on the AUC, 60% involved medications in different clinical areas, likely prescribed by different providers. Furthermore, the percentage of patients with EADGIs, in which multiple drugs or genes resulted in more severe interactions than the binary interactions typically flagged, ranged from 6% to 8.2%, depending on the minimum PIP score threshold. This once again highlights the need for a more global approach to PGx testing, including a comprehensive consideration of medications, not just those prescribed by one clinician in one clinical area, or for a single disease.

The utilization of PGx testing has also been shown to reduce the costs associated with medication management and any adverse reactions for the medications, which are highlighted in this study with the most major or contraindicated EADGIs. For example, implementation of PGx testing and management in opioid prescribing could save an estimated USD 14,000 per patient annually [37]. Further, *CYP2C19* genotype-guided antiplatelet therapy was found to likely reduce healthcare spending by an estimated USD 8525 in costs per patient [38]. Given the current Medicare reimbursement rate and assuming the need to test 50 patients to save one life, this equates to an estimated cost of USD 78,400 per life saved. Comparatively, assuming the lowest-cost statin was prescribed for five years, the cost to save one life would be at least USD 133,000–54,600 more than the cost to save one life with PGx-guided antiplatelet therapy [39]. Statin use is encouraged, with quality measures tied to financial impact, but there is not a single quality measure for genomics [40,41]. A recent model of quality-adjusted life year (QALY) analysis showed that a PGx strategy might be considered cost-effective with an incremental cost-effectiveness ratio (ICER) of EUR 60,000 per QALY when compared with no genetic testing for major depressive disorders; *CYP2D6* was cost-effective with an ICER of around EUR 47,000 per QALY [7,42]. A recent study demonstrated that incremental costs were USD 1646 lower with a gain of 0.04 QALYs when a multigene PGx panel was ordered instead of a single gene test for *CYP2C19* or *CYP2D6* [43]. These studies often reported findings for no more than a few genes and medications and did not account for the added lifetime value of the panel for optimizing other medications. Future studies, exploring the cost-effectiveness across all clinical areas and across multigene panels may demonstrate even greater savings and improved outcomes.

There is a financial and clinical case to be made for each of these examples, but when the implications are considered more broadly, the case for PGx panels becomes increasingly relevant. Since PGx test results are longitudinally constant, results can be preemptively utilized with appropriate CDSTs at the point of care for the lifetime of the patient. This will be further facilitated if Congressional efforts are successful in incentivizing updates to electronic health record systems with the goal of ensuring that healthcare providers are alerted to EADGIs when making prescribing decisions [1], similar to the existing drug and allergy alerts. Currently, Medicare covers PGx panel testing for patients, if they are taking medications or for a prescription that is under consideration that have a high-evidence impact, while many commercial insurance plans do not. Although polypharmacy is more common in older adults [44], the incorporation of CDSTs in PGx testing can provide early identification of patients at a high risk of adverse health outcomes at any age, and it can provide real-time evidence-based guidance on how to address EADGIs regardless of the clinical area.

### Limitations

Analysis was limited to data from the United States because the interpreted phenotype distribution used relied on North American data [15] and the cohort sizes in North American countries, outside of the United States, were small. The data set used in this study relied on test requisitions from physicians mainly in primary care, psychiatry, cardiology, and pain management and thus may not have necessarily captured all of the most prevalent interactions in the whole population. In addition, only reported data were analyzed, and data regarding how often results were acted upon were not available. PGx data for the minimum 3-gene panel are more established than those for genes in the larger panels; however, with expanded testing, more reliable data will be accumulated to advance evidence for other genes. Although patient race and ethnicity are known to impact prevalence of pharmacokinetic phenotypes, they are not currently accounted for in PIP scores. These data are collected at testing; however, self-reported data cannot be verified at this time, and phenotype prevalence data are not available for all tested genes. Efforts to incorporate racial and ethnic variation in future studies will be implemented as reference data improves, especially if use is expanded outside of North America. 

## 5. Conclusions and Future Work

This is the largest study to date to demonstrate the validity of a probability-driven algorithm that incorporates phenoconversion risk, specifically Invitae’s YouScript PIP score, to identify patients who would most likely benefit from PGx testing. The advantages of using this prescreening tool include automated incorporation of new and updated guidelines and a more efficient use of health care resources in order to help optimize medications across the entire patient care team. Prior studies have shown that the overall impact of using PGx testing and YouScript includes reduced hospitalizations, emergency department visits, and healthcare costs. Further validation of PIP scores must include the clinical impact in addition to quantifying comparability with testing results. Future work will also include additional validation in a broader population, randomized controlled trials, and coverage of more clinical areas and crossovers.

## Figures and Tables

**Figure 1 jpm-12-01972-f001:**
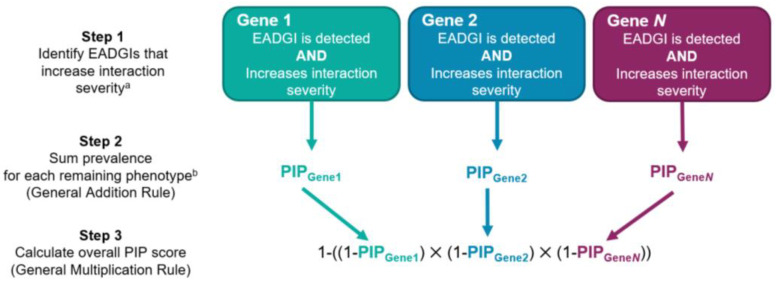
Workflow for calculating PIP scores. ^a^ The ability to detect the presence of an increased interaction severity for any EADGI or to identify any EADDGIs and EADGGIs is based on the YouScript knowledge base and patents [14]. ^b^ Only phenotypes such as ultrarapid metabolizer, intermediate metabolizer, or poor metabolizer with an increased interaction severity, are included in Step 2.

**Figure 2 jpm-12-01972-f002:**
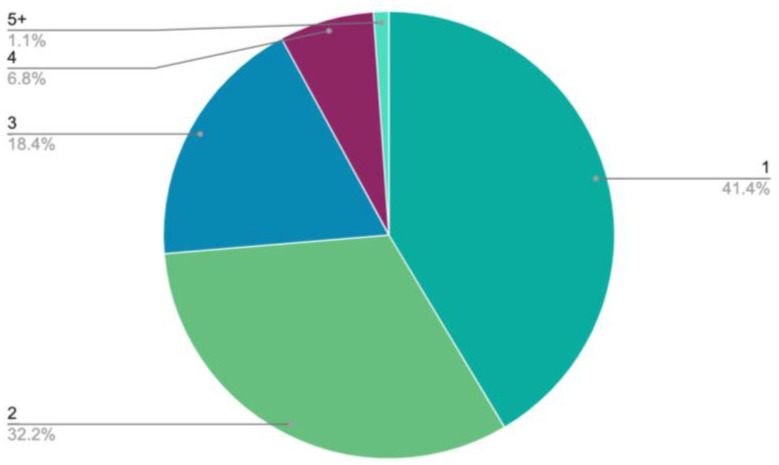
Proportion of patients with high-impact PGx medications by number of clinical areas impacted (*n* = 30,471).

**Figure 3 jpm-12-01972-f003:**
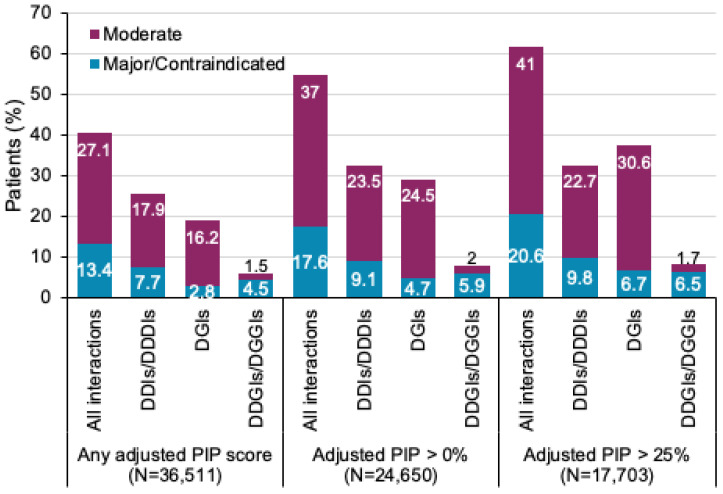
Comparative data on type and severity of interactions. If a patient had multiple identified interactions, only the interaction with the highest severity rating was considered in the frequency analysis per section. Abbreviations: drug–drug–drug interaction, DDDI; drug–drug interaction, DDI; drug–gene interaction, DGI; drug–drug–gene interaction, DDGI; drug–gene–gene interaction, DGGI; pharmacogenetic interaction probability, PIP.

**Table 1 jpm-12-01972-t001:** Patient characteristics.

Minimum No. of Genes in Panel	3 (All Study Patients)	5	14	25
Minimum Genes Included in Panel	*CYP2C19, CYP2C9, CYP2D6*	*CYP2C19, CYP2C9, CYP2D6, CYP3A4, CYP3A5*	*CYP2C19, CYP2C9, CYP2D6, CYP3A4, CYP3A5, CYP2B6, CYP4F2, SLCO1B1, TPMT, DPYD, HLA-B*57:01, IFNL3, UGT1A1, VKORC1*	*CYP2C19, CYP2C9, CYP2D6, CYP3A4, CYP3A5, ADRA2A, COMT, CYP1A2, CYP2B6, CYP4F2, DPYD, F2, F5, GRIK4, HLA-B*57:01, HTR2A, HTR2C, IFNL3, MTHFR, NAT2, OPRM1, SLCO1B1, TPMT, UGT1A1, VKORC1*
No. of Patients	36,511	28,613	3192	2068
Mean Age, years *	61	61	60	53
Age Range, years *^,^**	0–110	0–110	0–101	0–95
No. of Patients ≥ 65 years (%)	18,124 (49.6)	14,603 (51.0)	1721 (53.9)	758 (36.7)
Sex *, No. (%)				
Female	20,554 (56.3)	16,298 (57.0)	1674 (52.4)	1075 (52.0)
Male	14,360 (39.3)	10,995 (38.4)	1215 (38.1)	691 (33.4)
Unknown	1597 (4.4)	1320 (4.6)	303 (9.5)	302 (14.6)
Race *, No. (%)				
Black	2968 (8.1)	2478 (8.7)	44 (1.4)	44 (2.1)
Asian	387 (1.1)	346 (1.2)	14 (.4)	14 (0.7)
White	17,215 (47.2)	13,339 (46.6)	614 (19.2)	612 (29.6)
Hispanic	3413 (9.3)	2520 (8.8)	74 (2.3)	72 (3.5)
Jewish (Ashkenazi)	145 (0.4)	120 (0.4)	4 (0.1)	4 (0.2)
Unknown	12,383 (33.9)	9810 (34.3)	2442 (76.5)	1322 (63.9)
Mean No. of Medications per Patient (Range)	9.4 (1–62)	9.8 (1–62)	9.5 (1–62)	9.0 (1–62)
Mean No. of Variants or Variant Phenotypes per Patient	4.0	3.6	10.4	10.8
≥1 Variant or Variant Phenotype, %	96.9	97.7	100	100

* Age, sex, and race were self-identified on the test order form. ** “0” indicates patient either did not have a date of birth recorded (*n* = 4) or was <1 years old (*n* = 2).

**Table 2 jpm-12-01972-t002:** PIP scores and EADGIs detected.

Minimum No. of Genes Tested	No. of Patients	Mean PIP Score (25 Genes)	Mean Adjusted PIP Score * (to Minimum Genes Tested per Patient)	EADGI Rate ** (No. of Patients with at Least One EADGI)	NNT	*p*-Value (Mean Adjusted PIP Score vs. % EADGIs Detected)
3	36,511	26.4%	22.4%	22.4% (8174)	4.5	1.0000
5	28,613	27.5%	23.5%	23.4% (6707)	4.3	0.6895
14	3192	31.0%	30.9%	29.4% (937)	3.4	0.0667
25	2068	27.4%	27.3%	26.4% (545)	3.8	0.3583

Abbreviations: evidence-based, actionable drug–gene interaction, EADGI; number needed to test, NNT; pharmacogenetic interaction probability, PIP. * The adjusted PIP score was calculated using a Python script that created a small rounding difference. ** Determined by testing, based on FDA labeling or CPIC guidelines category A or B.

**Table 3 jpm-12-01972-t003:** EADGIs according to clinical area.

Clinical Area	No. (%) of Moderate EADGIs (*n* = 7360)	No. (%) of Major or Contraindicated EADGIs (*n* = 2444)	No. (%) of All EADGIs (Moderate, Major, Contraindicated) (*n* = 9804)
Behavioral Health	765 (10.4)	1106 (45.3)	1871 (19.1)
Cardiology	3230 (43.9)	877 (35.9)	4107 (41.9)
Pain Management	470 (6.4)	456 (18.7)	926 (9.4)
Hematology and Oncology	37 (0.5)	3 (0.1)	40 (0.4)
Infectious Disease	0	2 (0.1)	2 (0.0)
Gastroenterology	2634 (35.8)	0	2634 (26.9)
Urology	108 (1.5)	0	108 (1.1)
Transplant	8 (0.1)	0	8 (0.1)
Reproductive and Sexual Health	16 (0.2)	0	16 (0.2)
Neurology	42 (0.6)	0	42 (0.4)
Rheumatology	0	0	0
Endocrinology	0	0	0
Miscellaneous	50 (0.7)	0	50 (0.5)

Abbreviation: evidence-based, actionable drug–gene or drug–drug–gene interaction, EADGI.

**Table 4 jpm-12-01972-t004:** Ten most common EADDGIs (i.e., phenoconversion) with cumulative impact on AUC.

Affected Drug	Gene	Drug 2	AUC Change: Affected Drug + − Gene	AUC Change: Affected Drug + − Drug 2	Estimated AUC Change: Affected Drug + Gene + Drug 2
clopidogrel (metabolite)	CYP2C19 Intermediate Metabolizer	tramadol	−31–50%	−31–50%	−51–80%
citalopram	CYP2C19 Rapid Metabolizer	esomeprazole	−31–50%	26–75%	−0–30%
clopidogrel (metabolite)	CYP2C19 Intermediate Metabolizer	oxycodone	−31–50%	−31–50%	−51–80%
amitriptyline	CYP2D6 Intermediate Metabolizer	bupropion	26–75%	26–75%	76–200%
clopidogrel (metabolite)	CYP2C19 Intermediate Metabolizer	morphine	−31–50%	−31–50%	−51–80%
metoprolol	CYP2D6 Intermediate Metabolizer	dronedarone	76–200%	26–75%	>200%
clopidogrel (metabolite)	CYP2C19 Intermediate Metabolizer	hydrocodone	−31–50%	−31–50%	−51–80%
clopidogrel (metabolite)	CYP2C19 Poor Metabolizer	tramadol	−51–80%	−31–50%	−81–100%
amitriptyline	CYP2D6 Poor Metabolizer	bupropion	76–200%	26–75%	>200%
citalopram	CYP2C19 Rapid Metabolizer	fluvoxamine	−31–50%	26–75%	−0–30%

Abbreviation: area under the curve, AUC.

**Table 5 jpm-12-01972-t005:** Top five medications with EADGIs.

Drug (Clinical Area)	No. of Moderate EADGIs	Proportion of All EADGIs	Medication PIP Score	NNT	Rank among Top 200 Most Commonly Prescribed *
metoprolol (Cardiology)	2852	29.1%	48%	2.1	5
omeprazole (Gastroenterology)	1673	17.1%	29%	3.4	8
es(citalopram) (Behavioral Health)	1038	10.6%	32%	3.1	19, 30 (10 combined)
clopidogrel (Cardiology)	872	8.9%	29%	3.4	36
pantoprazole (Gastroenterology)	635	6.5%	29%	3.4	16

Abbreviation: evidence-based, actionable drug–gene or drug–drug–gene interaction, EADGI. * Drugs are ranked according to the 2019 ClinCalc DrugStats Database [19].

**Table 6 jpm-12-01972-t006:** Top five medications with major or contraindicated EADGIs.

Drug (Clinical Area)	No. of EADGIs	Proportion of Major or Contraindicated EADGIs	Medication PIP Score	NNT	Rank among Top 200 Most Commonly Prescribed *
es (citalopram) (Behavioral Health)	966	39.5%	32%	3.1	19, 30 (10 combined)
clopidogrel (Cardiology)	873	35.7%	29%	3.4	36
Tramadol (Pain Management)	305	12.5%	9%	11.1	35
codeine (Pain Management)	98	4.0%	9%	11.1	173
amitriptyline (Behavioral Health)	76	3.1%	50%	2.0	94

Abbreviations: evidence-based, actionable drug–gene or drug–drug–gene interaction, EADGI. * Drugs are ranked according to the 2019 ClinCalc DrugStats Database [19].

## Data Availability

The data analyzed in this study were obtained from Invitae’s internal databases. The data are proprietary and not available for public use but, under certain conditions, may be made available to editors and their approved auditors under a data-use agreement to confirm the findings of the current study. Further inquiries can be directed to Kristine Ashcraft at Invitae.

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
