# Peer review of "Validation of Pharmacogenomic Interaction Probability (PIP) Scores in Predicting Drug–Gene, Drug–Drug–Gene, and Drug–Gene–Gene Interaction Risks in a Large Patient Population"

_jpm, 2022, doi:10.3390/jpm12121972_

Round 1

Reviewer 1 Report

This retrospective study validates the accuracy of the Pharmacogenomic Interaction Probability (PIP) Scores in predicting patients who may have EADGIs and would benefit most from PGx testing. This new strategy contributes to reduce health care costs, enable patient access to personalized medicine, and ultimately improve drug efficacy and safety.

Minor comments:

1.      Lines 77-78: Regarding: “A complete list of drugs used to calculate PIP scores in the study, and their respective clinical areas, is available in the Supplementary Material (Table S1). “. I did not have the supplementary material available, which did not allow me to evaluate what is attached to this document.

2.      Line 85: Given the important usefulness of the proposed algorithm, I suggest that a figure (Flowchart or workflow) that summarizes it be designed and included in the item "2.2. YouScript Algorithm: Pharmacogenomic Interaction Probability". Its reproducibility should be facilitated in other types of patient populations .

3.      Line 203: Label the axis "y" of Figure 2. Comparative data on type and severity of interactions.

4.      Line 376: Given the limitations mentioned in the study, future studies should be suggested to expand patient cohorts with data from race and ethnicity, among others important variables in PIP scores analysis.

Reviewer 2 Report

The manuscript is clear and adequately presented. The authors reported the relevance of the PIP score for predicting EADGIs when compared to PGx tests. The results are interesting, they support the advantes of PGx implementation in the clinical practice, employing novelty options. I only have few comments:

- The number of 62 medications in Table 1, is correct?

- Please add a comma in the number 1,646 (line 345).

- The age is relevant for the presence of adverse events to drugs and polypharmacy. I can see a wide range in the age of patients included in the study (0-110). How PIP score consider this non-genetic factor?

- I find the Table 4 very important. May I suggest that the information could be also presented in a figure or a representative scheme?

- Table 6. I can see a congruence between number of EADGIs and 25-Gene PIP score, however for amitriptyline is different, with a PIP Score of 50% and 76 EADGIs. Authors could clarify this point. 

Round 2

Reviewer 1 Report

Accept in present form

Reviewer 2 Report

I appreciate the changes and clarifications made by the authors. The manuscript is suitable for publication.